# Refining Micron-Sized Grains to Nanoscale in Ni-Co Based Superalloy by Quasistatical Compressive Deformation at High Temperature

**Hui Xu [1,2,\*], Yanjun Guo [1], Jiangwei Wang [2] , Ze Li [1], Lu Wang [1], Xiaohui Li [1] and Zhefeng Zhang [3,\*]**

[1] Materials Research Center, Huadian Electric Power Research Institute Co., Ltd., Hangzhou 310030, China
[2] School of Materials Science and Engineering, Zhejiang University, Hangzhou 310027, China
[3] Shi-Changxu Innovation Center for Advanced Materials, Institute of Metal Research, Chinese Academy of Sciences, Shenyang 110016, China
[\*] Correspondence: hui-xu@chder.com (H.X.); zhfzhang@imr.ac.cn (Z.Z.)

**Abstract:** Compressive deformation was carried out in an Ni-Co-based superalloy with relatively low stacking fault energy (SFE) at 725 °C and a strain rate of $10^{-2}$ s$^{-1}$; the underlying micromechanisms were investigated under true compression strains varying from 0.1 to 1.0. It was found that dislocation slipping accompanied by stacking fault (SF) shearing dominated the compressive deformation under the strain of 0.1 and 0.2. As the strain increased to 0.3 and 0.4, microtwinning was activated and then interacted with dislocations, leading to the formation of dislocation tangles or blocky distorted region. When true strain was further increased to 0.6, abundant subgains (SGs) with polygonous shape appeared and then transformed into nanograins as true strain increased to 1.0. It is demonstrated that high strain and microtwinning are the prerequisites for the evolution of nanograins in the deformed Ni-Co-based superalloy. High strain can produce plentiful dislocations and distorted micro-sized SGs; then the microtwins sheared these distorted regions and refined the micro-sized SGs into nanoscale, which subsequently transformed into nanograins with further deformation.

**Keywords:** superalloy; nanograin; microtwin; dislocation; compression





## 1. Introduction

Ni-Co-based superalloys have been widely used in the discs of industrial gas turbines for their superior combination of service performance that includes excellent high-temperature strength, oxidation resistance, creep resistance, and fatigue properties [1]. In view of the unmatched mechanical properties of the present superalloys employed in gas turbines that demand a higher thrust–weight ratio and thermal efficiency, developing new superalloys with better service properties and designing advanced coatings with special protective effects are often seen as feasible approaches [2–4]. In recent years, a novel designing principle that strengthens superalloy at service temperatures and weakens superalloy at processing temperatures has been developed and applied in the modified Ni-Co-based superalloy by controlling its alloying element and stacking fault energy (SFE) [5–8]. SFE has a strong influence on the strengthening mechanisms, mechanical behaviors, and microstructural evolution of superalloys. Abundant studies have been carried out on the deformation behaviors of conventional superalloys with higher SFE during tensile or compressive tests. A consensus has been reached that dislocation motion dominates the deformation process at different conditions, and the precipitation strengthening of the γ′ phase is greatly affected by the mode of dislocation slip [9,10]. The modified Ni-Co-based superalloys possess low SFE due to their high electron hole concentration adding Co element [11], which exhibits distinct deformation behaviors with respect to the high-SFE superalloys. In the low-SFE superalloys, it was proved that microtwins (MTs) could be introduced during tensile deformation [12]. Furthermore, the ultimate tensile strength and

uniform elongation were improved synchronously because microtwinning was activated in superalloys with decreased SFE at 650 and 725 °C [13]. When deformation occurred at higher strains, nanograins (NGs) were found to be produced in the superalloys deformed at a high temperature and low strain rate, which resulted in higher flow stress [14]. However, the underlying mechanisms on the NG formation related to strain variation have not been clarified in these low-SFE superalloys based on the experimental analysis.

Moreover, grain refinement generally benefits the mechanical properties of bulk metal via strengthening and toughening the materials based on the Hall–Petch effect. In high-SFE metallic materials such as pure Ni, pure Fe, Cu, and Cu alloys, deformation at a low temperature and high strain rate can produce numerous NGs due to dislocation-mediated grain refining [15–18]. However, the nanocrystallization of modified Ni-Co-based superalloys shows them to be entirely different from these materials either in the deformation mechanisms or deformation conditions. For better understanding the underlying mechanisms of nanocrystallization with stain variation in Ni-Co-based superalloys, interrupted compressive tests were carried out for a newly developed Ni-Co-based superalloy in order to clarify the critical procedures for its grain refinement. NGs are beneficial for strengthening the Ni-Co-based superalloys, which can be utilized in the gradient structure of the superalloys that require surface strengthening.

## 2. Experimental

One kind of Ni-Co-based superalloy was selected as experimental materials with the chemical compositions of 14.6Cr-3.7Mo-20.5Co-1.9Al-5.7Ti-0.26Fe-0.03C-0.051Zr-Ni bal (wt.%). The master alloy ingots were first smelted in a 20 Kg vacuum arc furnace, and then homogenized heat treatment was performed to alleviate composition segregation. Then, round rods were cut from the treated ingots, which were subsequently encased and extruded into test bars with diameter of 35 mm at 1160 °C. Finally, the test bars were subjected to two-step solution treatments and two-step aging treatments for 1170 °C/4 h/AC + 1080 °C/4 h/AC + 845 °C/24 h/AC + 760 °C/16 h/AC (AC is air cooling), resulting in an average grain size of 124 μm determined by electron back-scattered diffraction analysis. Cylindrical specimens with dimensions of Φ 5 × 8 mm were machined by electrical discharge machining, followed by mechanical polishing to eliminate surface scratch. Compression tests were then carried out for the specimens at 725 °C and under the strain rate of $10^{-2}$ s$^{-1}$ using a Gleeble 3800 thermal simulation test machine with temperature control accuracy of ±1 °C, displacement measurement sensitivity of ±1%, and force measurement accuracy of ±1%, which were interrupted at different true strains ($\varepsilon$) varying from 0.1 to 1.0 in order to study the dependence of deformation microstructures on compression strains. The microstructure was characterized by an FEI Tecnai F20 transmission electron microscope (TEM) operated at 200 kV. TEM slices with thickness of 500 μm were cut from the middle part of the compressed samples along cross sections. The slices were further ground down to 50 μm and perforated by a twin-jet electro-polisher in a solution of 10% perchloric acid and 90% ethanol under conditions of 30~32 V and −22~−20 °C.

## 3. Results

The original microstructures were composed of coarse primary γ' precipitates, tiny secondary γ' precipitates, intergranular carbides, and γ matrix (Figure 1). Two kinds of γ' precipitate were coherently embedded in the face-centered cubic γ matrix. As compressive strain increased, phase constituents showed no apparent changes, while the deformed microstructures evolved continually. The deformation microstructures of the tested sample at lower strains are depicted in Figure 2. It could be found that deformation was dominated by dislocation motion since high density of dislocation tangles was introduced at the strain of 0.1 (Figure 2a); meanwhile, fine SF debris scattered both in the γ matrix and primary γ' precipitates. It could be found that SF propagation was inhibited by the phase interface of the γ' precipitates, leading to the independent growth of SFs, which were restrained either in the γ matrix or primary γ' precipitates (Figure 2b). As the strain increased to 0.2,

dislocation slip became more prevalent in order to accommodate larger compression strain; meanwhile, dislocation climb could be activated thermally when the slip of dislocations was hindered by the $\gamma'$ precipitates. Dislocation motion that included slip and climb continued to dominate the compressive deformation since high density of dislocation tangles also appeared in the $\gamma$ matrix (Figure 2c). The nucleation and propagation of SFs seemed to be enhanced in that multidirectional growth of SFs could be detected both in the $\gamma$ matrix and primary $\gamma'$ precipitates (Figure 2d). Meanwhile, the length of SFs generated from different slip systems increased obviously, and the SFs sheared with each other when they came across one another. The strengthening effect of the primary $\gamma'$ precipitates could be reflected by the fact that the $\gamma/\gamma'$ interfaces provided critical sites for hindering dislocation slip and SF propagation; it was found that scarce dislocations or SFs nucleated in the interior of the primary $\gamma'$ precipitates could pass across the $\gamma/\gamma'$ interfaces into the $\gamma$ matrix.

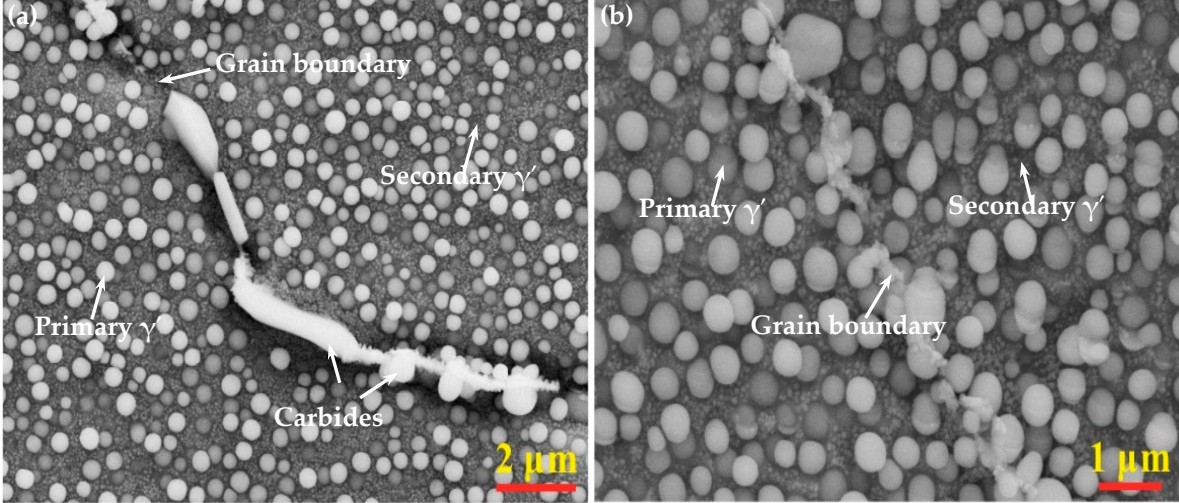

**Figure 1.** Typical microstructures of the samples observed by optical microscope: (**a**) original sample ($\varepsilon = 0$); (**b**) compressed sample ($\varepsilon = 1.0$).

The deformation microstructures of the tested sample at medium strains are depicted in Figure 3, which shows significant differences with respect to prior deformation at lower strains. When deformed at the strain of 0.3, numerous MT bundles were introduced that prevailed over the dislocation-controlled deformation and propagated parallelly with smaller spacing (Figure 3a). Meanwhile, SFs that extended along different directions were found to intersect with MTs, resulting in the formation of blocky obstacles, which would hinder dislocation motion and increase the density of dislocation tangle around these MTs. With the accumulation of dislocations due to the interaction between dislocations, SFs, and MTs, distorted stripes were introduced which had nearly straight boundaries that ran parallel to the MT boundaries, indicating the crucial role that MTs played on the formation of these distorted regions (Figure 3b). When deformed at the strain of 0.4, abundant distorted blocks emerged from the channels between the primary $\gamma'$ precipitates, and the size of the distorted regions grew obviously (Figure 3c). Simultaneously, the density of dislocation in the $\gamma$ matrix decreased remarkably due to long-term dislocation recovery, which was much more advantageous at higher strain. Under higher magnification, it could be found that these distorted blocks were sheared by fine SFs that propagated along two or more directions (Figure 3d). Furthermore, the interfaces of the primary $\gamma'$ precipitates and $\gamma$ matrix still had an impediment effect for SF shearing.

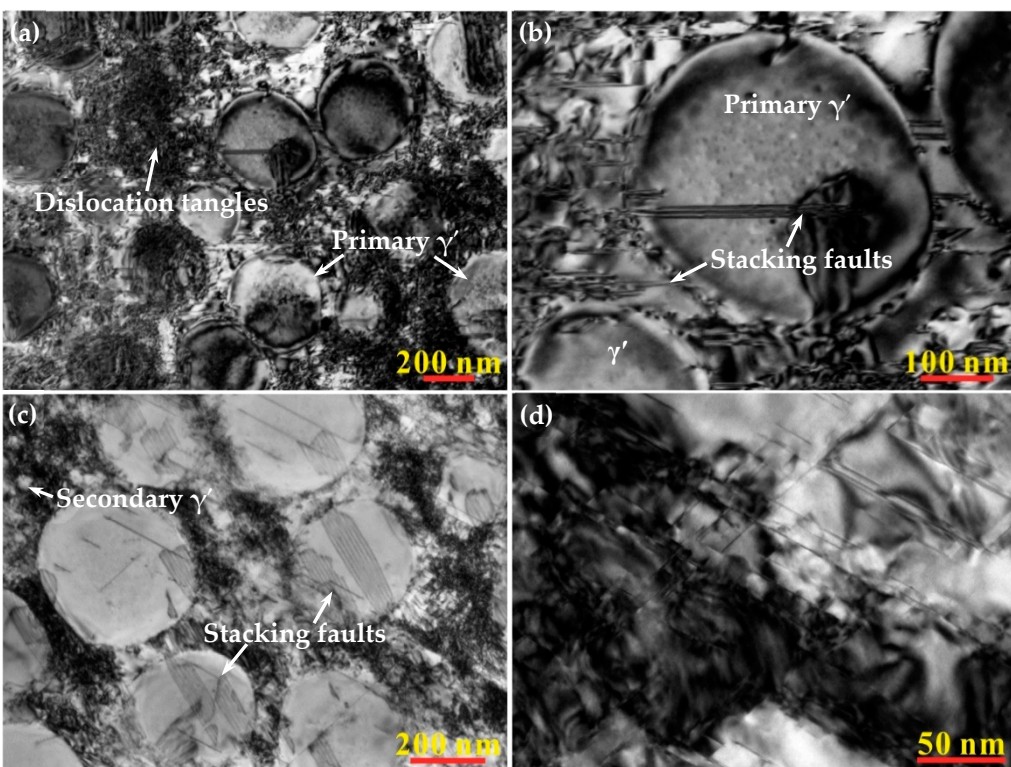

**Figure 2.** Typical microstructures of the compressed specimens at lower strains: (**a**) dislocation tangles, $\varepsilon$ = 0.1; (**b**) SF nucleation, $\varepsilon$ = 0.1; (**c**) SF propagation, $\varepsilon$ = 0.2; (**d**) interaction between dislocations and SFs, $\varepsilon$ = 0.2.

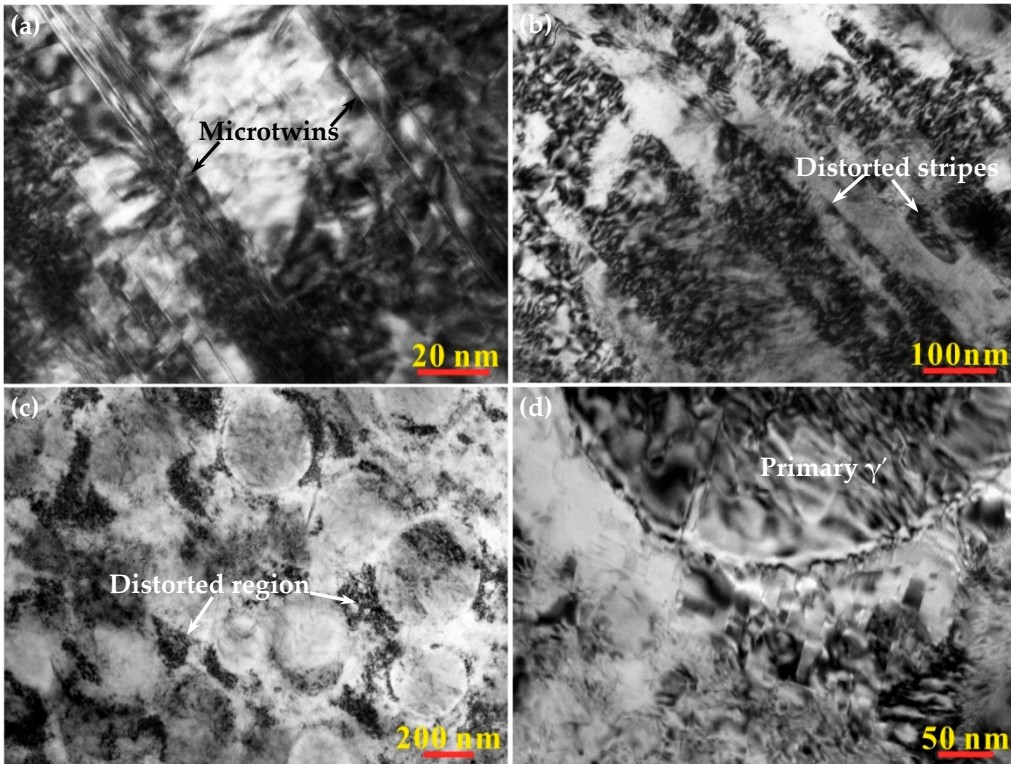

**Figure 3.** Typical microstructures of the compressed specimens at medium strains: (**a**) MT formation, $\varepsilon$ = 0.3; (**b**) strip-like distorted region, $\varepsilon$ = 0.3; (**c**) distorted blocks, $\varepsilon$ = 0.4; (**d**) the $\gamma/\gamma'$ interface, $\varepsilon$ = 0.4.

The deformation microstructures of the tested sample at higher strains were depicted in Figure 4. When deformed at the strain of 0.6, the primary $\gamma'$ precipitates were surrounded by numerous intersected SFs/MTs; these SFs/MTs seemed to penetrate the phase boundaries of primary $\gamma'$ precipitates on account of the continuous dislocation activities (Figure 4a). Apart from the numerous distorted regions, some SGs were found in the localized zone (Figure 4b). These SGs could be classified into two types, which included polygonal SGs with straight GBs and banded SGs that arranged parallelly. When the compressive test was carried out at the strain of 1.0, abundant SGs with polygonal shape formed that replaced the original distorted dark regions (Figure 4c). Besides these SGs, numerous nanograins (NGs) could be detected in the forms of equiaxial or irregular particles, which was verified by the presence of diffraction rings using selected area electron diffraction (Figure 4d). It was shown that SFs/MTs and SGs/NGs could hardly coexist since SFs/MTs were constantly absent from the regions that had plentiful SGs or NGs when deformation occurred at true strain of 0.6 and 1.0.

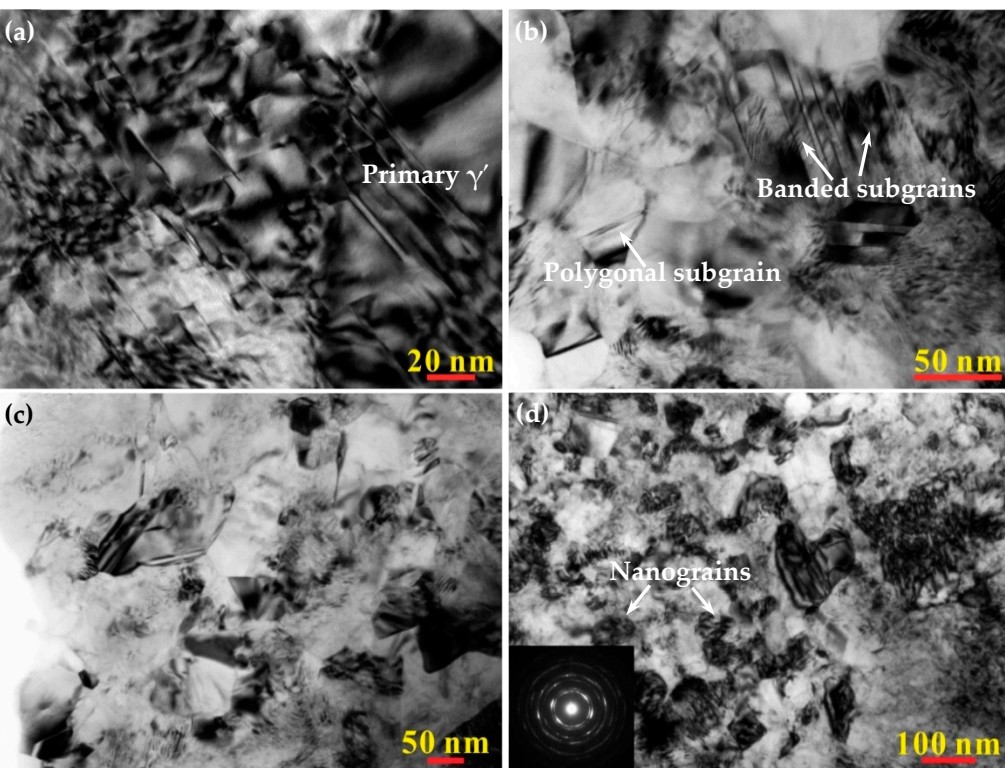

**Figure 4.** Typical microstructures of the compressed specimens at higher strains: (**a**) shearing of primary $\gamma'$ precipitates by SFs/MTs, $\varepsilon = 0.6$; (**b**) SG formation, $\varepsilon = 0.6$; (**c**) SG refinement, $\varepsilon = 1.0$; (**d**) NG formation, $\varepsilon = 1.0$.

## 4. Discussion

It has been well documented that dislocation slipping dominates the high-temperature deformation for superalloys with higher SFE, while SF shearing or microtwinning can be motivated in superalloys with lower SFE [19–21]. The tested superalloy has relatively lower SFE by adding 20% Co element in order to obtain a high concentration of electron hole. By interrupted compressive deformation investigation, it is interesting to find that the deformation micromechanisms of the tested superalloy underwent continual transformation with the implied strain level, which was dominated by dislocation slipping at lower strain, then SF/MT shearing at medium strain, and SG/NG forming at higher strain. It is worth noting that oxygen atoms have obvious impacts on the deformation behaviors of the tested samples since the deviation from stoichiometry and appearance of the oxygen anions can lead to some changes in the charge state of the cations, which in turn will greatly change



the electronic parameters [22,23]. That will seriously affect the practical application of the tested materials. It is well known that the complex transition metals and alloys easily allow the oxygen excess and/or deficit. In order to eliminate the adverse effects of oxygen atoms, the compression tests were carried out in a vacuum.

In this study, when the samples were deformed at lower strain and 725 °C, {111} <110> slip system was activated subsequently, then the a/2 <110> full dislocations dissociated into a/3 <112> Frank partial dislocation and a/6 <112> Shockley partial dislocation to facilitate the dislocation motion in face-centered cubic (FCC) metals [24]. Though the slipping dislocations could pass across fine secondary $\gamma'$ precipitates by a shearing mechanism, they would be impeded and piled up by larger primary $\gamma'$ precipitates unless dislocation climbing was activated. With the continuous slipping and climbing, most of the dislocations tangled around the $\gamma/\gamma'$ interfaces, which transformed into obstacles for the movement of other dislocations. Subsequently, slip of paired a/6 <112> Shockley partial dislocations gradually turned to prevail over the individual dislocation motion, which led to the initiation of SF debris in the $\gamma$ matrix and primary $\gamma'$ precipitates. Further deformation led to the multidirectional initiation and propagation of SFs and accelerated the formation of dense dislocation tangles.

When compressive deformation occurred at medium strains, microtwinning was activated that produced numerous MT bundles. MT is a kind of special deformation twin that has a thickness of 4–50 atom layers, which play an important role in the deformation mechanisms of low SFE superalloys. It is reported that MTs could synchronously improve the strength and plasticity by acting both as dislocation blockers and dislocation slip planes in Ni-Co-based superalloys during tensile tests [13]. It has been well documented that MTs were generally introduced by severe plastic deformation that applied high strain rate and deformation amount at a low temperature [25,26]. However, this study shows that MTs can be introduced in Ni-Co-based superalloy during compressive deformation at a lower strain rate of 0.01 s$^{-1}$ and a higher temperature of 725 °C, which is in accordance with other superalloys [27,28].

The underlying mechanisms for MT formation in the precipitation strengthening superalloys can be rationalized by a diffusion-controlled atom reordering theory [29,30]. This indicates that pseudo twins act as the critical prerequisite for MT formation, and that these are produced by the pairwise passage of four identical a/6 <112> Shockley partial dislocations along adjacent {111} slip planes. However, the pseudo twin has high-energy Al/Al nearest neighbor bonds in its complex stacking structure that are unstable in thermodynamics due to the relatively high anti-phase boundary (APB) energy in the LI$_2$-structured $\gamma'$ precipitate. When deformation is carried out at a high temperature (650–800 °C) and low strain rate, the transformation of pseudo twins into true MTs will be thermally activated by atom reordering that eliminates the high-energy Al/Al atom bonds after irreversible atom diffusion and exchange [30]. Once MTs are introduced, MT boundaries will act as the barriers for dislocation motion and result in dislocation tangle. Meanwhile, the intersection of MTs and SFs tends to aggravate the impediment effect for dislocation motion by locking dislocation slip. With the great increase in dislocation density at MT boundaries, distorted stripes are evolved from the highly tangled dislocations and detwinned MTs. Then, the distorted stripes will develop into larger distorted regions with irregular shape during further deformation.

Under higher strain level, polygonal and banded SGs appear initially, and then transform into equiaxial or irregular NGs as the further deformation is applied. When compressive deformation is carried out at the strain of 0.6, the distorted regions are enlarged to accommodate more dislocations. With the rotation of deformed grains, the orientation difference between the distorted regions and surrounding matrix increases gradually, which changes into SGs as a result of severe deformation. Then, microtwinning will be activated in the newly formed SGs when the orientation difference is high enough to hinder dislocation motion and MT shearing into the $\gamma$ matrix according to previous results [14]. Subsequently, the SGs are sheared and divided into finer SGs by MT cutting, which results

in the formation of polygonal and banded SGs. With the continual refinement effects of MTs in these SGs, the grain size of SGs decreases extremely and NGs will be introduced finally. It is shown that MT formation serves as the precursor for nanocrystallization in the tested superalloys since NGs are absent from the procedure without MTs and are introduced after the formation of MTs. It can be rationalized that initial deformation is necessary to supply numerous distorted regions for SGs formation, and microtwinning generally prevails over dislocation slipping when dislocation motion is impeded in the distorted regions. MTs play a critical role in subdividing and fragmenting the SGs, which serve as the precursor for NG generation.

## 5. Conclusions

In order to understand the nanocrystallization mechanisms of the newly developed Ni-Co-based superalloys, compressive tests were carried out at 725 °C and 0.01 s$^{-1}$ with different true strains. Three main conclusions can be drawn as follows:

(1) The deformation mechanisms of the tested superalloy evolve gradually with the applied compression strain, which in our study was controlled by dislocation slipping and SF shearing at lower strains, transformed into deformation microtwinning at medium strains, and then dominated by SG and NG formation at higher strains.

(2) A nanocrystallization approach is found in an Ni-Co-based superalloy with low SFE via compression deformation at a high temperature and low strain rate, which exhibits obviously differently from the high-SFE alloys that NG formation creates at a low temperature and high strain rate, revealing that the NG formation of the tested superalloy is a thermal activation-assisted process of microstructural evolution.

(3) The NG formation of the tested superalloy can be ascribed to high strain and microtwinning. High strain applied during compression tests produces plentiful dislocations and distorted micro-sized SGs; then, microtwinning resulting from low SFE will refine the micro-sized SGs into NGs with further deformation.

**Author Contributions:** Investigation—H.X., L.W., X.L.; writing—Original draft preparation, H.X.; writing—Review and editing, Z.L.; supervision Z.Z., J.W., Y.G. All authors have read and agreed to the published version of the manuscript.

**Funding:** This work was supported by the Key Project Program (No. CHDKJ200110) of China Huadian Corporation LTD, Beijing.

**Institutional Review Board Statement:** Not applicable.

**Informed Consent Statement:** Not applicable.

**Data Availability Statement:** The data of this study are available from the corresponding author upon reasonable request.

**Acknowledgments:** The authors are also grateful to the Material Service Behavior Division of Institute of Metal Research.

**Conflicts of Interest:** The authors declare no conflict of interest.

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
