# Peer review of "Refining Micron-Sized Grains to Nanoscale in Ni-Co Based Superalloy by Quasistatical Compressive Deformation at High Temperature"

_coatings, doi:10.3390/coatings13081325_

Round 1

Reviewer 1 Report

“Refining Micron-Sized Grains to Nanoscale in Ni-Co Based Superalloy by Compressive Deformation” by Hui Xu et al.

 I went through the paper very carefully and thoroughly. Authors interrupted compressive tests were carried out for a newly developed Ni-Co based superalloy in order to clarify the critical procedures for its grain refinement.

1- The impact of the paper on compressive tests is going be good. Also, the quality of the research work presented in the paper is also good. .

2-In general, ideas are well explained and understandable but, some tenses, linkers and grammar structures must be checked.

3-The authors should give the thickness and number of layers of all layers that they calculated as well as the diameters. Are these parameters obtained from an optimization process?

4. Authors should obtain the novelty of this manuscript compared to published results?

5. The authors should argue about the relevance of the temperature dependence of the coating.

6. The Introduction does not provide sufficient background. The introduction does not explain the major contributions and novelty of this work. The significance of the proposed solution has not been summed up.

7- The constructive discussions are missing. As mentioned earlier, authors must make a comparative analysis with other similar solutions and back up their claims on how the proposed solution can be considered as high performing compared to others.

8- The authors mentioned "As the strain increased to 0.2, dislocation motion that including slip and climb continued to dominate the compressive deformation since high density of dislocation tangles also appeared in γ matrix (Figure1c)" this report need evidence.

9-How their results will be affected if they include energy loss in layers.

10- The novelty of this work should be stated explicitly in the text of the manuscript so that readers can get it easily.

11- Authors should compare their results with the published data and different results.

12- There a lot of published papers in this field, authors should be explained the new in these results in sensors.

13- Authors mentioned” … In this study, when the samples were deformed at lower strain and 725℃, {111} 148 <110>slip system were activated subs. It is will be very useful if authors state the confirmation evidence.

14- Is the full dislocations can occurred 100%?

15- Authors should explain one or two applications to their work.

16- All figures, symbols, and units should be improved.

17- It seems the title need revision by authors to become more informative.

18- How this device can be stable with these kinds of materials.

19- The whole concept is now unclear to a reader what is the actual effect of the sensitivity of this device?

20- Are every term and structure in the proposed design should be clearly and correctly presented not to mislead the reader.

21- Is it possible to fabricate your results experimentally with good stability (you can state the time) by low cost method?

22-Finally, I recommend that the paper should be revised taking care of the above comments.

I wish to resend this paper after corrections and revise my comments

“Refining Micron-Sized Grains to Nanoscale in Ni-Co Based Superalloy by Compressive Deformation” by Hui Xu et al.

 I went through the paper very carefully and thoroughly. Authors interrupted compressive tests were carried out for a newly developed Ni-Co based superalloy in order to clarify the critical procedures for its grain refinement.

1- The impact of the paper on compressive tests is going be good. Also, the quality of the research work presented in the paper is also good. .

2-In general, ideas are well explained and understandable but, some tenses, linkers and grammar structures must be checked.

3-The authors should give the thickness and number of layers of all layers that they calculated as well as the diameters. Are these parameters obtained from an optimization process?

4. Authors should obtain the novelty of this manuscript compared to published results?

5. The authors should argue about the relevance of the temperature dependence of the coating.

6. The Introduction does not provide sufficient background. The introduction does not explain the major contributions and novelty of this work. The significance of the proposed solution has not been summed up.

7- The constructive discussions are missing. As mentioned earlier, authors must make a comparative analysis with other similar solutions and back up their claims on how the proposed solution can be considered as high performing compared to others.

8- The authors mentioned "As the strain increased to 0.2, dislocation motion that including slip and climb continued to dominate the compressive deformation since high density of dislocation tangles also appeared in γ matrix (Figure1c)" this report need evidence.

9-How their results will be affected if they include energy loss in layers.

10- The novelty of this work should be stated explicitly in the text of the manuscript so that readers can get it easily.

11- Authors should compare their results with the published data and different results.

12- There a lot of published papers in this field, authors should be explained the new in these results in sensors.

13- Authors mentioned” … In this study, when the samples were deformed at lower strain and 725℃, {111} 148 <110>slip system were activated subs. It is will be very useful if authors state the confirmation evidence.

14- Is the full dislocations can occurred 100%?

15- Authors should explain one or two applications to their work.

16- All figures, symbols, and units should be improved.

17- It seems the title need revision by authors to become more informative.

18- How this device can be stable with these kinds of materials.

19- The whole concept is now unclear to a reader what is the actual effect of the sensitivity of this device?

20- Are every term and structure in the proposed design should be clearly and correctly presented not to mislead the reader.

21- Is it possible to fabricate your results experimentally with good stability (you can state the time) by low cost method?

22-Finally, I recommend that the paper should be revised taking care of the above comments.

I wish to resend this paper after corrections and revise my comments

Author Response

Dear reviewer:

    Thanks for your constructive suggestions. I have completed the revisions. Please see the attachment.

Reviewer 2 Report

Referee Report

On the paper “ Refining micron-sized grains to nanoscale in Ni-Co based sup-eralloy by compressive deformation “ (coatings-2442057) by the authors Hui Xu, Yanjun Guo, Jiangwei Wang, Ze Li, Lu Wang, Xiaohui Li and Zhefeng Zhang submitted to the Coatings

This is interesting and useful paper. It reports the preparation and investigation of the structure and mechanical properties of the a Ni-Co based superalloy. Compressive deformation was carried out with relative low stacking fault energy at 725 °C and the strain rate of 10-2 s-1, the underlying micromechanisms were investigated under true compression strains varying from 0.1 to 1.0. It was found that dislocation slipping accompanied by stacking fault shearing dominated the compressive deformation under the strain of 0.1 and 0.2. As the strain increased to 0.3 and 0.4, microtwinning was activated and then interacted with dislocations, leading to the formation of dislocation tangles or blocky distorted region. When true strain was further increased to 0.6, abundant subgains with polygonous shape appeared, and then transformed into nanograins as true strain increased to 1.0. The obtained experimental results are reliable without any doubts. However, I have some questions and additions. I would like to note a few points to improve the paper before it can be published:

1.   The authors should give examples in 1. Introduction of the formation of thin films:

(1). A.L. Kozlovskiy, M.V. Zdorovets, Synthesis, structural, strength and corrosion properties of thin films of the type CuX (X = Bi, Mg, Ni), J. Mater. Sci.: Mater. Electron. 30 (2019) 11819-11832. https://doi.org/10.1007/s10854-019-01556-x.

2.   For metal and their alloys composite samples the stoichiometry is particularly important. The deviation from stoichiometry and appearance of the oxygen anions can lead to a change in the charge state of the cations, which in turn will greatly change the electronic parameters. That will seriously affect the practical application of the materials obtained. What is the oxygen stoichiometry of prepared samples? It is well known that the complex transition metal compounds easily allow the oxygen excess and/or deficit:

(2). S.V. Trukhanov, A.V. Trukhanov, A.N. Vasiliev, A.M. Balagurov, H. Szymczak, Magnetic state of the structural separated anion-deficient La0.70Sr0.30MnO2.85 manganite, J. Exp. Theor. Phys. 113 (2011) 819-825. https://doi.org/10.1134/S1063776111130127.

(3). A. Kozlovskiy, K. Egizbek, M.V. Zdorovets, M. Ibragimova, A. Shumskaya, A.A. Rogachev, Z.V. Ignatovich, K. Kadyrzhanov, Evaluation of the efficiency of detection and capture of manganese in aqueous solutions of FeCeOx nanocomposites doped with Nb2O5, Sensors 20 (2020) 4851. https://doi.org/10.3390/s20174851.

This should be discussed in 3. Results and 4. Discussions.

3.   The authors should mention in 1. Introduction such experimental methods of non-destructive testing and determination of microstresses in materials as X-ray or/and neutron diffraction:

(4). D.I. Shlimas, A.L. Kozlovskiy, M.V. Zdorovets, Study of the formation effect of the cubic phase of LiTiO2 on the structural, optical, and mechanical properties of LixTixO3 ceramics with different contents of the X component, J. Mater. Sci.: Mater. Electron. 32 (2021) 7410-7422. https://doi.org/10.1007/s10854-021-05454-z.

4.   The proposed 4 papers should be inserted in References.

The paper should be sent to me for the second analysis after the major revisions.

Minor editing of English language required

Author Response

(The authors gave the same response as above.)

Reviewer 3 Report

The article is devoted to the study of compressive deformation of cobalt-nickel alloy. The article is very small, which is why perhaps many points are not reflected in the article and raise additional questions.

1. The introduction is very short and contains only 9 reference links. The essence of the studies performed earlier is not reflected and it is not clear what caused this work and what is its relevance. What is the purpose of this research?

2. The sentence in lines 53-55, in my opinion, should be moved to the Introduction.

3. The investigated alloy in line 52 is indicated with an error. Cobalt is listed twice (14.6 Co and 20.5 Co). It turns out that readers do not know what alloy was investigated in this work.

4. Sample processing modes (lines 57-63) are better presented in the article in the form of a graph, so that the reader can understand it better. Now it is difficult to perceive this information.

5. Why was the compression carried out at a temperature of 725 °C?

6. In what part of the sample after compression were studies carried out? When compressing cylindrical samples, the deformation is distributed unevenly in the metal. The difference in deformations can be very significant.

7. In my opinion, it is necessary to bring the microstructure from the optical microscope of the original sample and samples after deformation. How was the average grain size of 124 µm determined (line 61)?

8. "The original microstructures are composed of coarse primary γ′ precipitations, tiny secondary γ′ precipitations, intergranular carbides and γ matrix." How did you define it? What is the chemical composition of γ′ and what is the composition of γ?

  9. Conclusions on the work are of a general nature and do not contain new knowledge. What is the novelty of your work?

10. If you did research on the Gleeble 3800, why didn't you show curves "true stress-true strain". This would be of interest to other researchers.

Author Response

(The authors gave the same response as above.)

Round 2

Reviewer 1 Report

Authors responded adequately

Author Response

Dear reviewer,

    Thanks very much for your good comments for the revised manuscript. You gave me constructive suggestions to improve this manuscript, and I have learned a lot from the revision process. The cited references have been revised again according to another reviewer's advice that adding two more related references in the part of Discussions. Wish you everything goes well! 

Yours sincerely,

 Hui Xu   

Reviewer 2 Report

Referee Report

On the paper “ Refining micron-sized grains to nanoscale in Ni-Co based sup-eralloy by compressive deformation “ (coatings-2442057-v2) by the authors Hui Xu, Yanjun Guo, Jiangwei Wang, Ze Li, Lu Wang, Xiaohui Li and Zhefeng Zhang submitted to the Coatings

Although some changes have been implemented, significant additional effort is still required to improve the paper. The authors should be more attentive and scrupulous to the suggestions and additions of the reviewer in order to achieve the desired result promptly:

1.    For metal and their alloys composite samples the stoichiometry is particularly important. The deviation from stoichiometry and appearance of the oxygen anions can lead to a change in the charge state of the cations, which in turn will greatly change the electronic parameters. That will seriously affect the practical application of the materials obtained. What is the oxygen stoichiometry of prepared samples? It is well known that the complex transition metal compounds easily allow the oxygen excess and/or deficit:

(1). S.V. Trukhanov, A.V. Trukhanov, A.N. Vasiliev, A.M. Balagurov, H. Szymczak, Magnetic state of the structural separated anion-deficient La0.70Sr0.30MnO2.85 manganite, J. Exp. Theor. Phys. 113 (2011) 819-825. https://doi.org/10.1134/S1063776111130127.

(2). A. Kozlovskiy, K. Egizbek, M.V. Zdorovets, M. Ibragimova, A. Shumskaya, A.A. Rogachev, Z.V. Ignatovich, K. Kadyrzhanov, Evaluation of the efficiency of detection and capture of manganese in aqueous solutions of FeCeOx nanocomposites doped with Nb2O5, Sensors 20 (2020) 4851. https://doi.org/10.3390/s20174851.

This should be discussed in 3. Results and 4. Discussions.

2.    The proposed 2 papers should be inserted in References.

The paper should be sent to me for the second analysis after the major revisions.

Minor editing of English language required

Reviewer 3 Report

The authors have corrected the paper. I recommend accepting for publication in this version.

Author Response

Dear reviewer,

    Thanks very much for your good comments for the revised manuscript. You gave me constructive suggestions to improve this manuscript, and I have learned a lot from the revision process. Wish you everything goes well!